# Educators as Health and Education Mediators for African Asylum Seekers in Israel

**DOI:** 10.3390/ijerph19095200

**Published:** 2022-04-25

**Authors:** Dolly Eliyahu-Levi

**Affiliations:** Levinsky College of Education, Tel Aviv 6937808, Israel; doly.levi@levinsky.ac.il; Tel.: +972-506944452

**Keywords:** health mediators, asylum seeker, personal relationships, educators

## Abstract

Israel is not isolated from the global migration process. It is required to provide a medical, educational, and socio-cultural response to the integration of tens of thousands of African asylum seekers. This qualitative-phenomenological study collected data from 15 educators as a primary source and learned about their actions to mediate health and educational issues for African asylum seekers. The findings reveal four categories: (1) a healthy lifestyle; (2) emotional-behavioral; (3) learning disabilities and special needs; (4) diseases, vaccines, and medical treatments. It seems that educators are forced to take on roles traditionally entrusted to the state, and they have become agents of socialization who mediate between parents and the Israeli health and education system through personal relationships and individual conversations. This study reveals a dual reality: on the one hand, African asylum seekers experience alienation, exclusion, and violence; on the other hand, they gain a positive point of view when parents see the educators as loyal partners and sources of knowledge who can be consulted to receive help in routine times and during the coronavirus pandemic, a time in which they lost their livelihood, health insurance, and ability to understand the new rules of the lockdowns.

## 1. Introduction

Israel is not isolated from the global migration process, and it is required to provide a medical, educational, and socio-cultural response to the integration of tens of thousands of African asylum seekers. A very significant proportion of them reside illegally in the country, in difficult living conditions on the social margins, while suffering from general, occupational, and family instability and experiencing a sense of alienation and social non-belonging. Asylum seekers experience hostility and racism in the public space from an early age, including comments on skin color, cursing, spitting, insults, and abusive graffiti on walls in residential neighborhoods and schools [1,2,3].

According to the Population and Immigration Authority, there are currently about 50,000 African asylum seekers living in Israel, most of them from Eritrea and Sudan. About 8000 children (most of them born in Israel) study throughout the country, of whom about 2300 study in separate schools in south Tel Aviv.

The Israeli government pursues a policy of exclusion and does not take responsibility for asylum seekers. Moreover, in recent years the Israeli government has made various attempts to reduce the number of asylum seekers and make their lives difficult: relocating them to open detention facilities, providing financial assistance for voluntary return, resettlement in third countries, and more [4,5].

In such a reality, municipal entities have had to take on the direct care of the new residents, allocating funds for this purpose from the municipal budget. They receive most of the services from third sector organizations and community organizations. In addition, the presence of asylum seekers in the southern neighborhoods of Tel Aviv has changed their social fabric and even created conflicts with the local population [6].

In Israel, according to the ‘Compulsory Education Law’, education is compulsory and free for all children from the age of 3 to 17, regardless of their citizenship. Refugee children have access to the primary and secondary education system under the same conditions as Israeli citizens. However, for most of them the situation is quite different: They are perceived as strangers and are located at the bottom of the social ladder [7,8]. As such, they are one of the weakest groups in Israel, excluded from many services and living in economic distress. Many children suffer from significant gaps and delays in learning, and the schools’ ability to provide a comprehensive and effective response is limited. The result is that children do not receive academic support and are unable to complete high school successfully. This fact is a blow to the young refugees’ dreams of a better future [9,10,11,12].

It is agreed that the subjects of this study—educators in schools that integrate children from families of asylum seekers—are among the significant factors for the children. These educators develop on their own initiative culturally adapted teaching practices for integration, adjustment, inclusion, and reduction in gaps. They act out of respect and sensitivity to the culture and tradition of the country of origin and play the role of intercultural mediators who promote socialization processes in order to facilitate coping with a lack of information, bureaucratic barriers, and educational conflicts [13,14,15,16]. 

Studies [17,18,19] indicate that educators’ mediation actions may establish health and educational knowledge among parents and children and help foster a continuous and supportive trust-based communication system. Moreover, in order to foster effective educational work, to understand that all the teaching and interactions with students take place in a cultural context that is neither neutral or accidental, and to maintain open, positive, and lasting communication, educators are required to possess intercultural competences that include knowledge, skills, and understanding of diverse cultural frameworks [20,21]. 

From the global aspect, in view of the demographic changes and globalization processes, and out of a desire to enrich the theoretical and practical knowledge regarding the integration of African asylum seekers in Israeli sociocultural spaces, I chose to focus on health and education mediation processes occurring between African asylum seeker parents and their children’s educators—educators who have a responsibility to develop a climate of integration between new and old groups and majority and minority groups and to encourage their involvement in the public space.

This article examines the perceptions and actions of educators in Tel Aviv’s elementary schools and high schools, who act as mediators for asylum seekers in Israel. The research question is as follows: What are the health and educational mediation actions of educators among asylum-seeking families?

At the beginning of the article, I review the community of asylum seekers in Israel from two aspects: medical and educational and the role of educators as mediators. I then present the research method and analysis of the findings regarding the mediation actions relating to health and educational issues. At the end of the article, a summary and conclusions are presented.

### 1.1. The Medical Aspect

According to the law regulating the provision of health services in Israel, most asylum seekers are not entitled to health services because they do not have a civil status that entitles them to resident rights. They are only eligible to receive emergency medical care in hospitals. In Tel Aviv, there are the following: (1) a designated clinic for the statusless, (2) a Levinsky clinic for sexually transmitted diseases and AIDS, and (3) a refugee clinic operated by volunteers [22]. In addition, minors from families of asylum seekers and refugees born in Israel and whose parents stayed in Israel for six months are entitled to subsidized medical insurance from the Ministry of Health. However, due to financial difficulties, some parents cannot insure their children, and they are left without health insurance. Data from the Ministry of Health show that more than 20% of refugee minors lack health insurance [23].

The coronavirus pandemic has exacerbated the situation of asylum seekers in Israel, a weakened and excluded population without any social and health support network, which is at greater risk of infection. The isolation and elimination of jobs meant that 80% of the asylum seekers found themselves in a severe crisis, without work, a livelihood, health insurance, entitlement to unemployment benefits, and severance pay. They find it very difficult to recover from the effects of the coronavirus crisis and return to their daily routine, which was difficult and complex to begin with [24].

### 1.2. The Educational Aspect

The Government of Israel has determined that following the Compulsory Education Law, children from families of asylum seekers and refugees without status residing in Israel will study in the public education system from age three [4,25,26]. However, children are not entitled to pedagogical and social support such as allocating school hours to improve the acquisition of the language of instruction, didactic diagnoses, or participation in programs to treat complex learning disabilities, language difficulties, or emotional problems [27].

In this complex reality, the challenge facing educators working in the intercultural arena with children from African asylum-seeking families is to understand the socio-cultural differences and find effective and appropriate educational solutions for parents and children. This requires developing intercultural competence among educators as a central tool for creating cooperation between them and the parents while establishing ways of communication and trust [20,28]. Educators with intercultural competence are able to develop a climate of integration between cultures within the educational framework, fulfill the role of mediators to promote new medical, educational, and cultural knowledge, find appropriate solutions to their diverse problems, and prevent discrimination of asylum seekers. They can also make it easier for parents and children to deal with barriers and conflicts [29,30,31]. 

Studies in education indicate [7,10,32] that educators play an essential and significant role concerning children from families of asylum seekers. The children describe a positive relationship of trust between them and the teachers. According to them, teachers are a significant factor influencing their motivation, self-confidence, and academic success.

Educators in multicultural and multilingual settings who teach children from migrant, refugee, and asylum-seeker families should have intercultural competence and plan classroom lessons based on principles of culture-relevant pedagogy. Educators also play the role of language and culture mediators and find themselves explaining educational and health issues to children and parents. Their main role is to cultivate ways of communicating and mediating messages that are not clear to asylum seekers [28,29].. Researchers in education and migration [33,34,35] show that personal relationships, empathy, personal exposure, and listening to personal issues help children and parents deal with social tensions and promote a positive attitude toward the general society.

According to Berhanu [36] and Sisneros [37], intercultural mediation encourages members of different cultural groups to strengthen their “belonging” and identification with the dominant group. It does not mean abandoning the culture of the country of origin in favor of the dominant culture; rather, it means facilitating the process of forming a multicultural society that gives a sense of space and allows the various groups within it to adapt to the system of values, norms, and behaviors of the majority.

The purpose of this study is to examine and describe the actions of educators in an elementary school to mediate health and educational issues for African asylum seekers in Israel.

## 2. Methods

This qualitative-phenomenological study emphasizes the subjective interpretation that people give to the socio-cultural reality they find themselves in and draws its data from the natural array. The qualitative paradigm opens up new possibilities for observing human behavior, understanding it, and creating knowledge. The role of the researcher is to investigate a phenomenon, find its meaning, and interpret it [38].

The phenomenological approach deals with the study of the essence of human experience. The phenomenon is the object of inquiry, and the researcher strives to explore the meaning of the experience for the subjects in their daily lives. Phenomenological research deals with individuals’ experiences and how they perceive them. Researchers in this approach do not take reality for granted but examine the nature of things according to the contexts and circumstances in which they observe. A phenomenon perceived from one point of view will not be the same as that perceived from another point of view. Therefore, the essence of phenomenological research is in creating theoretical generalizations [39,40].

Moreover, this approach makes it possible to collect data from educators as a primary source and learn about their perceptions, actions, feelings, unique experiences, and the interactions in which they are involved. As a result, research findings may assist in constructing educational programs that promote the integration of marginalized minority populations in a multilingual and multicultural society [41].

Fifteen educators participated in the study: three men and twelve women. They all teach in multicultural and multilingual primary schools, where children from immigrant families, refugees, and asylum seekers comprise the majority of the student body. The institutional vision of these schools incorporates a pluralistic conception rooted in the values of equal rights and tolerance, respect for other languages, and acceptance of difference without preconditions.

All educators are aged 25–48, with five years or more of teaching experience. They all revealed an educational concept that emphasizes the importance of personal contact with children and their parents, despite language difficulties and cultural gaps. One of the educators taught for three years in the United States in a culturally, religiously, and nationally diverse school. He gained experience teaching a foreign language and contacting parents from cultural and national minority groups.

The study was conducted in the southern neighborhoods of Tel Aviv, where most asylum seekers live, alongside the migrant population. The presence of African asylum seekers has changed the urban space. As part of the change, for example, they are seen in various places in the city, riding bicycles, cleaning streets, walking by the sea, gardening, and being employed in restaurants. Their presence is also sometimes revealed in protest measures such as demonstrations, handing out leaflets, or strikes.

Surveys [42,43,44] show that veteran Israeli residents of Tel Aviv feel an economic, social, and national threat from asylum seekers, even from those perceived as “real refugees.” Moreover, the majority of the Israeli public fears that asylum seekers will increase the level of crime in society and become a burden on welfare systems, and among many there is growing support for tightening immigration policy and reducing social rights [45,46].

Due to fears of the development of social exclusion, which could negatively affect the conduct of the city of Tel Aviv on various levels, the municipality established in 1999 an assistance and information center for the community of foreigners called “Mesila”, serving migrant workers, asylum seekers, and refugees. All activities are funded by the municipality and donations from private individuals. The assistance center provides information services in all areas of life as well as a response to primary needs such as food, clothing, etc. in situations of stress [47]..

The study data were collected from personal interviews with 15 participants. Each interview lasted about an hour and was conducted at the participants’ workplace. The topics discussed in these interviews revolved around the actions of educators to mediate health and educational issues for African asylum seekers. The data were analyzed through content analysis that focused on what participants said rather than how things were said. According to Shkedi [48], content analysis is a window into the inner experience of the interviewee. The analysis focuses on the words and descriptions of the educators as reflecting their actions, perceptions, feelings, beliefs, and knowledge. Moreover, Krippendorff [49] claims that content analysis allows the description of data and the drawing of valid conclusions for their broad context.

I strictly followed the necessary ethical rules: maintaining the anonymity and confidentiality of the respondents and the data, avoiding abusive questions, and giving educators a choice whether to participate in the study or not.

## 3. Results

Examining the actions of primary school educators in mediating health and educational issues for African asylum seekers revealed four content categories of mediation: (1) a healthy lifestyle: nutrition and hygiene; (2) emotional-behavioral; (3) learning disabilities and special needs; (4) diseases, vaccines, and medical treatments.

### 3.1. A Healthy Lifestyle: Nutrition and Hygiene

Here is an example that illustrates the educators’ actions as mediators of healthy lifestyle issues through a dialogue with parents that includes explanations of the importance of a daily shower, brushing teeth, and trimming nails:


*“On the subject of cleanliness there is neglect: they do not wash their hair, nor do they trim their nails. I think we need to comment, explain to children and parents. If you explain, you can see the change. I try to explain to parents in simple words, in pictures, in body gestures, as they do not speak Hebrew, how important it is for the children to shower and brush their teeth. We even did a “body cleaning” day and gave them toothbrushes and toothpaste. Sometimes the kids come a few days in a row with the same clothes. They have a shower at home, but the parents arrive late, the children are already asleep, without showering, and get up in the morning for school. Parents do not really have time for children; they work crazy hours, so I also explain to children that it is important to shower even if mom and dad are not home (M).”*


M. seems to be aware of the gaps in norms of behavior between what is acceptable in the child’s home and what is acceptable and desirable in Israeli society, the receiving society. It is important for her that the parents understand the message, so she tries to be creative and mediates the message through pictures, simple words, and body movements. Her words “if you explain you can see the change” prove that she believes that if she makes the parents listen, understand, and believe her, change can occur. M. also tries to explain the fact that the children do not shower every day, claiming that the parents work until late, are busy with existential troubles, and do not always have the time, strength, or patience to take care of the children’s hygiene as well.

B. describes an ongoing dialogue with the parents on the issue of eating sweets. Out of concern for children’s nutrition, she tries to encourage children and parents to change their eating habits:


*“From morning and throughout the day they eat sweets, gummy candies, lollipops. The parents buy them sweets because they think it calms the children. When the children cry, get angry, argue, the parents think that candy is a solution. I talked to the parents, I explained to them, but nothing helps, they continue to buy sweets. I’m not sure their eating habits can be changed. It can be difficult to convince them to eat more fruits and vegetables because their price is very high (B).”*


B. tries to influence the issue of sweets but is unsuccessful. Parents seem to be aware of the harm to their children’s health but are unable to reduce their consumption of sweets, perhaps because of their cheap price compared to the high price of fruits and vegetables, or perhaps because they have difficulty setting boundaries for children, explaining, and educating out of parental authority, thus choosing sweets as an easy solution. Perhaps the parents know how difficult their children’s lives have been, and they are allowing them to eat sweets to try to make them happy.

The healthy eating issue also came up in interviews with other educators. Here is another example, where A. shows concern about eating habits at home:


*“The school has delicious and varied sandwiches for breakfast. I at least know that here they sit at the table eating a sandwich and vegetables, because I do not know what is going on in the house, in terms of setting boundaries, hygiene, or how many sweets they eat (A).”*


This example also illustrates the teachers’ concern about what is happening in the children’s homes from various aspects: education, health, and nutrition. The children from the refugee families seem to benefit from a government-funded feeding program, which is intended, among other things, to improve the nutrition of children from vulnerable groups for whom healthy food is not available. From the point of view of the educator, there may be a gap in eating habits between what is acceptable at school and what is familiar at home.

### 3.2. Emotional-Behavioral

These two educators describe actions aimed at helping children and parents deal with social, emotional, and behavioral difficulties, with frustration and social processes accompanied by expressions of violence and racism:


*“I have been teaching at the school for seven years, and I do not believe there will be a change. I’m not optimistic. It should be understood that there is a very big difference between “real life” in the neighborhood and “protected life” in school. There is chaos, confrontation, and violence in the neighborhood. The Israeli residents curse the Africans, shout at them to return to where they came from, call them names such as “monkeys” “garbage”, “blacks”. There is a lot of frustration, verbal violence, and sometimes even beatings. The children live in daily risk. At school we make great efforts to enable them to live a normal life. I want to allow children to dream in a safe environment, to teach them to take responsibility, to strengthen their self-esteem, their motivation. The best way is through personal conversations. I make time for these meetings, and sometimes stay in the afternoon as well. Through the personal relationship I manage to touch them personally, to really listen to them. The kids know I’m here for them, they can trust me (L).”*



*“I have an open communication channel with the parents, and yet their starting point is so unequal, and they live in such difficult conditions, the parents work 12–14 h a day, for them they try to do the maximum, but they cannot support the children. To empower parents emotionally and help them deal with frustration and helplessness, we set up a forum that meets once a month. The forum is attended by parents, representatives from the municipality, and teachers. We sit together, the parents raise issues that exist in the neighborhood, at school, and think together about a solution. The forum gives parents a sense that there is someone who cares, who listens to them, who tries to help (T).”*


Both educators, L. and T., act as mediators in a multicultural environment, caring for the children’s mental health and trying to alleviate emotional frustration through personal dialogue with the children and with African asylum-seeking parents. The mediating discourse takes place not from a paternalistic position, superiority, and power of a dominant culture, but as a dialogue at “eye level”, while establishing trust, closeness, and openness [29].. Researchers [33,34,35,50] have found that close relationships based on listening, respect for diverse beliefs, empathy, personal exposure, and concern can help address challenges and tensions while reducing the gap between the dominant society and the minority group.

### 3.3. Learning Disabilities and Special Needs

In the educators’ reports, I found evidence of educators’ actions aimed at finding appropriate solutions to learning disabilities, communication problems, developmental delays, and special needs of children, while mediating the knowledge to the parents. Pedagogical activities included individual learning, increasing parental involvement in the learning process, and informal diagnoses.


*“Children and parents do not receive support for learning from an official office such as the municipality or government—only volunteers. As far as the state is concerned, they are not recognized, they do not have an identity card, they are just a number. It all depends on the initiative and goodwill of the teachers. I take the children to individual meetings, teach them the new words, practice until they start reading in Hebrew. In case I find more complex difficulties, I try to diagnose the problem. Luckily, I also took special education courses. I have to work differentially in class (R).”*


The Israeli education system expects educators in classes with children from African asylum-seeker families to bear educational responsibility and to lead all students to meet the required achievements and standards. That is, educators are required to have a broad-based pedagogical knowledge base, good will, and the ability to adapt to the linguistic teaching-learning methods of the children in the classroom without official municipal or political support. Studies [51,52,53,54] show that classroom educators respond on a practical level to differences and take a concrete pedagogical approach while addressing teaching methods, cognitive abilities, and language adaptation for cultural minorities.

One of the ways in which educators address learning difficulties and special needs is through personal-differential teaching that allows each child to learn regardless of their abilities. There is research evidence for the effectiveness of differential teaching in achieving learning goals, using methods such as individual work, peer teaching, and providing children with a choice of topics and teaching styles [55]. Moreover, the personal relationship between educator and child may increase children’s self-confidence and self-efficacy and help deal with stress [55,56,57]. Gieras [58] also believes that personal connection, an emotionally supportive discourse, and positive reinforcements encourage the children and increase the understanding that their success depends on them and their investment.


*“Without parental involvement we will not be able to advance the children in school—and this is an almost impossible task. To increase the partnership, we have a group on WhatsApp, I upload pictures of the kids, write what we learned, what the tasks are. Some parents understand Hebrew a little better, and they translate and pass on the information to the other parents in Tigrinya (F).”*


F. reveals a pedagogical perception according to which cooperation with parents is of great importance, especially when it comes to parents from a minority group on the social margins, whose socio-behavioral norms are different from those accepted in the receiving educational setting. It seems that asylum-seeking parents lack the knowledge, skills, and social support to deal with the difficulties posed by the receiving society, and educators understand that in order to fulfill their role as mediators, they must strengthen cooperation with parents. For example, the encounter with the African asylum-seeker mothers was an opportunity for direct mediation that enriched the mothers with new knowledge about engaging in learning processes without using violence toward the children, even in situations of tension and argument.

Studies examining the relationship between parental involvement and student achievement from minority groups [17,19,59,60,61] show that there is a positive relationship between the two. It is therefore important that the educational setting develop and expand the range of activities that encourage parental involvement.

### 3.4. Diseases, Vaccines, and Medical Treatments

Educators reported personal conversations with parents when the children were sick, not feeling well, or injured. They explained to parents where to go and what to do. They mediate information about local clinics where they can receive treatment, even though they do not have health insurance:


*“I check the children if they have a fever, mycosis, skin sores, or other medical problems. If there is a problem, I call the parents via WhatsApp. They are in no hurry to come and pick up the child from school. The parents take the child to the doctor and call me because they do not understand what the doctor wrote. Sometimes I write a letter to a doctor, and sometimes I have to explain where the clinic is for someone who does not have health insurance.*
*The next day the child comes to class because they do not understand the concept of “staying at home” (B).”*


The educator helps parents deal with medical issues while drawing their attention to such issues. She mediates information written by the doctor and accepted norms of behavior such as that a child stays home when he is sick, and also helps them to overcome language difficulties in the face of medical factors. The mediation reveals to the asylum-seeker parents the way in which the Israeli health system is run and encourages them to take responsibility for the children’s health, to take time to take care of the children, and to be with them at home until they recover. Mediation seems to reduce the gaps and help the asylum seekers to learn and understand that they can receive medical care even under living conditions in which they are not considered equal citizens in the receiving country.

Reality forces H. to play a new role as a “health-educational mediator.” Health information is not accessible to helpless parents, many of whom were left without health insurance during the coronavirus pandemic, after losing their jobs. After all, health insurance is activated only when asylum seekers work for an employer. Moreover, they are not entitled to state support benefits, and many of them suffered from food insecurity during the pandemic:


*“No one explains to them how the health care system works in Israel. What should they do? When? Who do you turn to in an emergency? Where is there a volunteer clinic? Do you have to buy medical insurance? I feel I am the source of knowledge, the “address” for them. When they have a problem—not necessarily related to education or school—they turn to me. During the corona period, it intensified. The educators were the source of information during the lockdowns, vaccinations, and the issuance of a green pass. Many asylum seekers have difficulty receiving green passes or immunization certificates because the government system does not have all of their details. The educators were a listening ear to them and helped them even when the schools were closed. The school administration also approached several organizations to provide them with food, baby products and more. Each educator examined the needs of her families (H).”*


## 4. Discussion

The findings of the study revealed four content categories of mediation: (1) a healthy lifestyle; (2) emotional-behavioral; (3) learning disabilities and special needs; (4) diseases, vaccines, and medical treatments. The educators reported personal relationships and individual conversations with the parents in all four categories. They provided them with relevant health-educational information about options for receiving medical care even for those without insurance and explanations of accepted social norms and behaviors in the receiving society. The parents saw the educators as an “address” for every question and problem in all areas of life.

It also shows that in the reality of the Israeli government’s exclusion policy regarding the integration of African asylum seekers into society [4], educators are forced to take on roles traditionally entrusted to the state and become health and education mediators for African asylum seekers. Thus, educators have become agents of socialization, and their role takes on a new meaning as a personal-social factor mediating between parents and the Israeli health and education system, at their formal and informal levels; in doing so, the educators go beyond their formal roles.

Educators find themselves in a space that deepens educational inequality while facing many difficulties resulting from children’s lack of background knowledge and community characteristics and lack of professional guidance. Despite this, they act as mediators and express a conception of human pedagogy that follows cultural responsiveness [31,62,63,64], i.e., responding out of respect for the other culture [65,66] and maintaining personal communication based on trust with the parents—listening, explaining, translating, advising, encouraging, and supporting.

The personal relationships led parents to perceive the educators as “honest” mediators with intercultural competence that includes knowledge of the other culture, communication skills, the ability to resolve conflicts between groups, empowering social justice, reducing gaps, and dealing with stereotypes and prejudices [20,67]. 

Moreover, in the local Israeli context, the findings indicate that the mediation process helps to reduce tensions between the home system and the general social system outside it. In examining the global context, Gratton, Gutmann, and Skop [68] studied families and children in migration processes around the world and found that these tensions have an impact on family functioning, child integration in the education system, and family integration in the receiving society [69]..

The findings also reveal the difficult situation of asylum seekers throughout the coronavirus pandemic, during which the medical mediation of the educators to the parents was more significant and important than ever. With the loss of livelihood, asylum seekers also lost their only way to receive medical care in the community. Additionally, during the pandemic, by virtue of the Patient Rights Act, they could only receive treatment in emergency rooms. The problems were many and complex: (1) the medical information was not accessible to asylum seekers; (2) there was a fear that asylum seekers would arrive in the emergency rooms contrary to the guidelines of the Ministry of Health; (3) asylum seekers suffering from serious or chronic illnesses—some of which were life-threatening—did not receive appropriate treatment; (4) the lack of medical insurance did not allow asylum seekers to be tested for the coronavirus [56].

In a reality where the employment crisis led to an economic and humanitarian crisis in all areas of life of the asylum-seeking community, and they were left without a state support network, educators played an important role as a source of knowledge in everything related to vaccines, the rules of conduct in lockdowns, what is allowed and what is forbidden and the possibility of receiving a “green pass”. Moreover, the educators also acted to prevent a humanitarian catastrophe. They collaborated with volunteers and non-profit organizations, kept in touch with parents, found out what they were lacking, and took care to provide food, basic products, diapers, medicines, and more.

These findings may have implications for the international arena, as the mediation activities described can be a working model for promoting and integrating other immigrant populations elsewhere in the world. The mediation activities position educators as a significant personal and ethical anchor for asylum-seeking parents and their children. These activities are important because they place the African asylum seekers at the center of the educational endeavor and as a top priority of the educational setting; they strive to address the challenges of integration between groups and give asylum seekers an opportunity to integrate, reduce gaps, and possibly even live independent lives.

## 5. Conclusions

The term “multiculturalism” carries sociological significance and refers to a situation in which different cultures exist within a single political framework. A multicultural approach is a model for educational action that includes recognition of cultural diversity, the right of individuals to preserve their cultural uniqueness while ensuring their full access to rights, resources, and opportunities in society, all with the aim of addressing social (in)equality and promoting integration and inclusion [70,71].

This study reveals the authentic voice of primary school educators: perceptions, experiences, and mediating actions in multicultural educational frameworks that incorporate children from African asylum-seeking families, who are different from the general society in race, religion, skin color, culture, ethnicity, and culture.

This study reveals a dual reality: on the one hand, African asylum seekers face bureaucratic, social, economic, and social challenges. They experience alienation, exclusion, and violence that amplify the difficulties in the process of social integration. Additionally, it seems that many Israeli citizens do not show openness and tolerance towards them. Moreover, the coronavirus crisis has exposed government policy and the injustice it causes to asylum seekers, as the closure of the economy and the cessation of work propelled the population of asylum seekers to the brink of an economic and mental crisis as they face economic insecurity and inability to pay rent [24].

On the other hand, a positive point of view is revealed when parents see the educators as loyal partners, as a source of knowledge, as figures who can be consulted to receive help in routine and times of crisis. Educators revealed an educational perception in which they understand the other and his or her point of view, identify difficulties, and work to reduce information and knowledge gaps. They do not seek to change asylum seekers’ lifestyles and do not show a preference for Israeli culture over the culture of their country of origin; rather, they enable the preservation of asylum seekers’ tradition while deepening their familiarity with the education system, health system, and accepted norms of behavior in the receiving society.

The most prominent pedagogical principles adopted by the educators in the mediation process were the establishment of personal relationships and the strengthening of cooperation. The personal connection between the educators and the parents grew through continuous communication and mutual trust that enabled effective functioning and cooperation, finding solutions in a good spirit, and even organizing properly for emergencies [72]. Mediation through personal relationships helped parents correctly interpret the Israeli educational, health, and cultural context, to receive health care even in an environment that often-revealed hostility, resistance, or lack of acceptance [25]. In addition, the mediation process was accompanied by mediation of language gaps and the understanding that each of the parties had different norms of behavior and values. According to Arzubiaga, Noguerón, and Sullivan [17], in order to overcome the difficulties, prior recognition of the worldview of each of the parties is required. Such prior acquaintance may prevent frustrations and disappointments.

The findings of this study are inconsistent with other studies that examined the relationship between educators and refugee parents and asylum seekers [73,74,75], which found that most parents do not trust the educational staff and therefore avoid contact, personal or family exposure, and prefer to maintain limited contact with the educational establishment. It is possible that in the Israeli context, African asylum-seeker parents did not find a sympathetic ear in other urban or political sources, so that educators who expressed empathy, good will, and acceptance were perceived as trustworthy and reliable in the eyes of the parents.

From a practical aspect, the findings of the study highlight the importance of expanding the direct and personal dialogue between educators and parents from asylum-seeking families or with parents from other minority groups. It is important and necessary that in the process of teacher training, students cultivate empathy, sensitivity to others, intercultural competence, and develop skills of socio-cultural mediation that deepens the acquaintance between the parties and avoids expressions of racism, exclusion, and power. Such training will raise educators who can deal with complex situations while addressing academic, emotional, and cultural differences related to the learners’ abilities.

## Data Availability

Not applicable.

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
