# Peer review of "Educators as Health and Education Mediators for African Asylum Seekers in Israel"

_ijerph, 2022, doi:10.3390/ijerph19095200_

Round 1
Reviewer 1 Report
Overall, I found this manuscript persuasive, and the methodological approach provides a great foundation for future, larger-N studies. This is a high-quality qualitative manuscript that is written in a way that is accessible for a non-qualitative audience.
I think the one portion of the paper that I struggled with buying the argument was the passage on page 5 which examined the Teacher trying to change eating habits, specifically the child eating too many sweets. I'm wondering how the authors are able to claim that the teacher can tell the parents know they are unhealthy, or if there are any additional quotes which could better support the suggestions below the quote about the parents' motivations for not changing the behavior. Could it not also be that as Asylum Seekers, the parents know how difficult their children's lives have been, and are allowing them to eat sweets to try to make them happy? I think the author should explore this section further. That feels nit-picky, but it is the weakest area in an otherwise very strong manuscript.
Author Response
Dear Reviewer (1)
I want to thank you for the professional reference and all the critical comments.
I addressed your argument about children's eating habits, especially eating sweets:
- I made sure that the interpretation in the body of the article would be carefully written, including reservations - "it seems that ..." - "it is possible that ..." so that the interpretation presented is not unequivocal.
- I have added another example that illustrates the educators' concern regarding the eating habits in the children's home.
- I added an explanation in the text - the parents may allow them to eat sweets to try to make the children happy.
- Prior to resubmission, the manuscript will undergo professional language editing in English
Thank you!
Reviewer 2 Report
Review report – ijerph-1666873-peer-review-v1
The manuscript deals with an interesting topic that of the educators acting as health and education mediators for the integration of African asylum seekers in Israel. The empirical exercise is useful yet it is weekly related to the available knowledge in the field while a more clear and insightful presentation and analysis of the knowledge provided by the interviews is needed. Also, the structure of arguments and the overall presentation of the text need considerable improvements.
Below please see a summary of changes that would allow for better communicating the study’s main argument and importance.
- A first major concern relates to the study’s lack of a clear presentation of ‘hypotheses’ (‘propositions’) as stemming from the available knowledge in the field. Discussion of available knowledge in the field is scattered in parts 1 and 4 (introduction and the discussion part) of the text but the reader does not get a clear understanding of what is to be expected and why. I would suggest that a background knowledge part is added to the text with a clear presentation of arguments as drawn from the international literature and in a subsection as related to Israel. Then ‘hypotheses’ or anticipated phenomena will be clearer to describe.
- In light of point 1 above and the overall restructuring of the text I would suggest that Part 1 should be made stronger by adding some relevant data, e.g. indicative numbers of refugees etc. Also, data on policy initiatives and probably funds allocated to such measures will give a picture of the financial burden that is caused for the country and its efforts to accommodate migrants, refugees, etc. The policy context is relevant for the analysis of the study’s results and policy implications.
- In part 2 of the study a table giving the basic info on participants (gender, age position in the school) is typically reported along with the initials of the respondent or any other identification code. Also please provide a clear description of the case study area where the interviews were conducted and the area’s profile with regard to hosting migrants and refugees.
- The discussion and conclusion parts of the study should be rewritten in light of the above changes.
- The text needs to be corrected for syntax errors throughout. Also please check for repetitions etc.
Author Response
Dear Reviewer (2)
Thank you for the professional reference and all the critical comments. These comments focused the article on research questions and deepened the academic discussion.
All additions and corrections are in different font colors.
Regarding the comment regarding background knowledge of the text
- I expanded the introduction chapter and based it on international studies.
- I explained the broad Israeli context of the research described in the article: numerical data, Israeli government policy, the Israeli education system, the role of educators, and the choice of topic.
- I also described the course of writing the article, and now it is clear to the reader what is expected of him, what he is about to read.
Regarding the note - strengthening Part 1
- I added data on the numbers of asylum seekers in Israel.
- Unfortunately, there are no economic data on the extent of investment in projects made for the families of asylum seekers in Israel. Most of the activity is voluntary, and the funds are donations.
Regarding the note - strengthening Part 2
In the second chapter, the methodology chapter, I added a detailed description of the area where the interviews with the teachers were conducted - South Tel Aviv. I also added information about the attitude of the residents towards asylum seekers.
- In the discussion chapter, I added a reference to a global perspective.
- Before submitting the manuscript, it will undergo professional language editing in English.
Thank you!
Reviewer 3 Report
This manuscript aims to explore the role of educators in African asylum seekers. The research design is good, and the findings is important to the development of Israel. However, some improvements should be considered in this manuscript.
1. The author need to establish the importance of this research topic from a globel perspective.
2. There are many mistakes in English expression。
3. The literature reivew can not support your research design.
4. The author need to make more on the empirical analysis and results.
Author Response
Dear Reviewer (3)
Thank you for the professional reference and all the critical comments. These comments focused the article on research questions and deepened the academic discussion.
All additions and corrections are in different font colors.
- Before submitting the manuscript, it will undergo professional language editing in English.
- In the discussion chapter, I added a reference to a global perspective
- I added a detailed description of the area where the interviews were conducted with the teachers - South Tel Aviv. I also added information on the residents' attitudes towards asylum seekers. This information helps in understanding the chapter of the findings.
Thank you!
Reviewer 4 Report
The objective of this article is to provide insights into educators as agents of socialization who mediate between parents and the Israeli health and education system through personal relationships and individual conversations, particularly in the case of tens of thousands of African asylum seekers. See abstract, lines 7-20.
In the 1. Introduction, the authors illustrate the situation of tens of thousands of African asylum seekers, who are in difficult social conditions and experience hostility and racism in the public space from an early age. See lines 25-34.
Sub-paragraph 1.1 The Medical Aspect, illustrates which difficulties asylum seekers encounter in Israel since they are not entitled to health services and are only eligible to receive emergency medical care in hospitals. Situation in Tel Aviv sees a couple of clinics for stateless and sexually transmitted diseases ill people and one refugee clinic operates by volunteers. Notwithstanding Israel allows to financial support through subsidized medical insurances from the Health Ministry to minors from families of asylum seekers and refugees born in the Country, parents due to a difficult financial situation fail the same to insure their children. See lines 37-47.
A worsening of these conditions has been provoked by the financial shock encountered by Israel during the pandemics. See lines 48-55.
Sub-paragraph 1.2 The Educational Aspect, illustrates how children from families of asylum seekers and refugees without status residing in 58 Israel will study in the public education system from three, but they are not entitled to pedagogical and social support. This involves the issues of serious matters on their participation in programs and subsequent learning and emotional abilities.
Educators then play a pivotal role for both establishing ways of communication and trust to overpass barriers and conflicts and developing a climate of integration between cultures. See lines 57-76.
Sub-paragraph 1.3 Educators as mediators, describes some of these important features for Educators, such as (i) intercultural competence, (ii) classroom work’s planning on the basis of culture-relevant pedagogy, (iii) empathy, (iv) personal exposure, (v) ability at listening.
All these duties can be exemplified in the so-called intercultural mediation, which encourages members of different cultural groups to strengthen their “belonging” and identification 93 with the dominant group. Again, the purpose of this paper is showed at line 99-100, that is to examine and describe the actions of educators in an elementary school to mediate health and educational issues for for African asylum seekers in Israel.
Section 2. Methods shows the method of the authors, that is a qualitative-phenomenological study emphasizes the subjective interpretation that people give to the socio-cultural reality they find themselves in and draws its data 103 from the natural array. See lines 102-105.
Lines 108 forwards deals with the explanation of the phenomenological approach.
In particular, this approach for the study means to collect data from fifteen teachers (three men and twelve women) in multicultural and multilingual primary schools, where mainly children from immigrant families, refugees, and asylum seekers and learn about their perceptions, actions, feelings, unique experiences, and the interactions in which they are involved. See lines 117-121.
Each interview lasts about an hour in the participants' workplace.
Some hints about “content analysis” are furnished at lines 136-142.
In their Section 3. Results, the authors group their findings about the study into four content categories of mediation:
- a healthy lifestyle: nutrition and hygiene,
- emotional-behavioral,
- learning disabilities and special needs,
- diseases, vaccines, and medical treatments. See lines 147-151.
Subsequent sub-paragraph 3.1 a healthy lifestyle: nutrition and hygiene, offers excerpts from teachers’ interviews, demonstrating (i) the willing of these teachers to be creative and mediate messages through pictures, simple words, and body movements, for example about self-hygiene, and (ii) the encountering of difficult setting boundaries for asylum seekers’ children. See lines 153-187.
Subsequent sub-paragraph 3.2 emotional-behavioral, offers excerpts from teachers’ interviews, demonstrating (i) the need to teach asylum seekers’ children to take responsibility, to strengthen their self-esteem, their motivation, because they often encounter frustration and racism in the neighborhood, and (ii) the chance to open forums with parents to overpass this situation. See lines 189-210.
Lines 211-221 are an enlisting of how challenges and tensions can be overpassed through mediating discourse.
Subsequent sub-paragraph 3.3 learning disabilities and special needs, offers excerpts from teachers’ interviews, demonstrating (i) the need for teachers to have a broad-based pedagogical knowledge base, good will, and the ability to adapt to the linguistic teaching-learning methods of the children in the classroom without the support of a municipal or political official, and (ii) the need of parental involvement. See lines 223-271.
Subsequent sub-paragraph 3.4 Diseases, vaccines, and medical treatments, offers excerpts from teachers’ interviews, demonstrating (i) educators have to know how to mediate information on local clinics and health treatments and insurance, and (ii) the ability to teach how the Israeli health systems is run to let asylum seekers understand that they can and how to receive medical care. See lines 273-311.
Section 4. Discussion, recapitulates main findings in the paper, especially the concern Educators must keep on health matters. See lines 313-364.
Section 5. Conclusions, states the personal view of the authors about “multiculturalism,” especially to the light of the delicate duties of the Israeli educators, who lever on personal relationships with their pupils and on the strengthening of cooperation between families of asylum seekers and public authorities. See lines 366-414.
CHANGE REQUEST:
- Please in the numbered list at lines 149-151 put “Disease” in small initial letter as all the other three items.
Instead in the four sub-paragraphs’ titles at lines 152, 188, 222, and 272, put all the first words’ initial letters in Capital letter as already done only at title line 272,
- Dashes here and there in the paper including “inferential” discourses should be longer than they are and without spaces in-between, see by instance lines 213, 229, 360, 370, 384. Please check together with the Assistant Editor at MDPI,
Example: “On the other hand—a positive point of view is revealed…”.
- Line 314, same as point 1,
- Please put in bold also title Conclusions at line 365,
- Please put sub-paragraph 1 The Medical Aspect, line 36, sub-paragraph 1.2 The Educational Aspect, line 56, and sub-paragraph 1.3 Educators as mediators, line 76, with the same formatting as sub-paragraphs 3.1 at line 152 and following 3.2, 3.3, and 3.4,
- Please put “Trust” at line 409 with small initial letter.
With Kind Regards,
References:
Moss D, Gutzeit Z, Mishori R, et al Ensuring migrants’ right to health? Case of undocumented children in Israel BMJ Paediatrics Open 2019;3:e000490. doi: 10.1136/bmjpo-2019-000490
Author Response
Dear Reviewer (4)
Thank you for the professional reference and all the critical comments.
All additions and corrections are in different font colors.
- I corrected all the errors mentioned.
- Before submitting the manuscript, it will undergo professional language editing in English.
Regarding the Introduction
- I added data on the numbers of asylum seekers in Israel.
- Unfortunately, there are no economic data on the extent of investment in projects made for the families of asylum seekers in Israel. Most of the activity is voluntary, and the funds are donations.
Regarding the methodology chapter
- I added a detailed description of the area where the interviews with the teachers were conducted - South Tel Aviv.
- I also added information about the attitude of the residents towards asylum seekers.
Thank you!
Round 2
Reviewer 2 Report
The authors have made every effort to accommodate the suggested changes.
I would suggest that minor syntax errors are corrected in this version of the manuscript.
Author Response
Dear Reviewer (2)
Thank you for your professional comments regarding the revised manuscript.
I forwarded the article for further reading by an English language editor.
Thank you!
Reviewer 3 Report
The revised manuscript can provide some interesting findings to readers, and the findings of this work is important for the field of education.
Author Response
Dear Reviewer (3)
Thank you for your professional comments regarding the revised manuscript.
I forwarded the article for further reading by an English language editor.
Thank you!